# Modulation of the Gut Microbiota by Tomato Flours Obtained after Conventional and Ohmic Heating Extraction and Its Prebiotic Properties [note 1]

**DOI:** 10.3390/foods12091920

**Published:** 2023-05-08

**Authors:** Marta C. Coelho, Célia Costa, Dalila Roupar, Sara Silva, A. Sebastião Rodrigues, José A. Teixeira, Manuela E. Pintado

**Affiliations:** 1CBQF—Centro de Biotecnologia e Química Fina—Laboratório Associado, Escola Superior de Biotecnologia, Universidade Católica Portuguesa, Rua Diogo Botelho 1327, 4169-005 Porto, Portugal; 2CEB—Centre of Biological Engineering, University of Minho, 4710-057 Braga, Portugal; 3Centre for Toxicogenomics and Human Health (ToxOmics), Genetics, Oncology and Human Toxicology, NOVA Medical School|Faculdade de Ciências Médicas, Universidade Nova de Lisboa, Campo dos Mártires da Pátria, 130, 1169-056 Lisbon, Portugal

**Keywords:** gut microbiota, short-chain fatty acids, prebiotic

## Abstract

Several studies have supported the positive functional health effects of both prebiotics and probiotics on gut microbiota. Among these, the selective growth of beneficial bacteria due to the use of prebiotics and bioactive compounds as an energy and carbon source is critical to promote the development of healthy microbiota within the human gut. The present work aimed to assess the fermentability of tomato flour obtained after ohmic (SFOH) and conventional (SFCONV) extraction of phenolic compounds and carotenoids as well as their potential impact upon specific microbiota groups. To accomplish this, the attained bagasse flour was submitted to an in vitro simulation of gastrointestinal digestion before its potential fermentability and impact upon gut microbiota (using an in vitro fecal fermentation model). Different impacts on the probiotic strains studied were observed for SFCONV promoting the *B. animalis* growth, while SFOH promoted the *B. longum,* probably based on the different carbohydrate profiles of the flours. Overall, the flours used were capable of functioning as a direct substrate to support potential prebiotic growth for *Bifidus longum*. The fecal fermentation model results showed the highest Bacteroidetes growth with SFOH and the highest values of *Bacteroides* with SFCONV. A correlation between microorganisms’ growth and short-chain fatty acids was also found. This by-product seems to promote beneficial effects on microbiota flora and could be a potential prebiotic ingredient, although more extensive in vivo trials would be necessary to confirm this.

## 1. Introduction

The gut microbiota arrangement depends on individual intrinsic factors (e.g., age, ethnicity, genetic markers) and environmental factors (e.g., geographic area, lifestyle, diet, and drugs) [1,2], whereas the host intestine provides the necessary environmental conditions for the bacteria therein to survive and reproduce. The gut microbiota modulates various physical functions (e.g., nutrient processing and digestion, immune cell growth and immune response, and immunity towards pathogens, among others), hence representing a mutualistic relationship [2].

Most bacterial fermentation happens in the proximal colon, where there is higher substrate accessibility. Toward the distal colon, the convenience of substrates falls, and the recovery of available food reduces both substrate and microbial products’ distribution. The fermentation results in the production of short-chain fatty acids (SCFAs) together with gas (CO_2_ and H_2_) [3]. These compound molecules are mainly generated in the large intestine by gut microbiota fermentation of carbohydrates that had escaped digestion and absorption in the small intestine, although non-digested proteins or peptides are also essential upstream compounds for their production [4,5]. 

Bioactive compound sources are mainly found in plants, such as tomato fruit. While also used for fresh consumption, tomatoes are primarily used for processing into juice, pulp and sauces, hence originating many by-products whose potential valorization is still scarce [6,7]. A full and integrated recovery from tomato by-products with zero residues, in a context of a circular economy, could be used as a strategy. For instance, the final solid extraction by-product can be dried under controlled conditions, resulting in flour with a high fibre content combined with bonded bioactive compounds, such as phenolic compounds and carotenoids [8,9,10]. As such, the resulting material could be used with an ingredient that, given its characteristics, could have interesting biological potential, particularly in the modulation of the gut microbiota [11,12,13]. In addition, different extraction techniques have been tested to valorize these by-products, including “green techniques” like ohmic (OH) processing, which obtain bioactive extracts with significant differences from extracts obtained by conventional extraction [14]. 

Still, even though diet arrangement has been demonstrated to have a modulating effect on gut microbial communities, knowledge of the effects exerted by particular foods in driving gut microbial variety is limited, hence hampering their optimal use.

Therefore, the present work aimed to characterize the prebiotic potential of two tomato flours obtained after ohmic (SFOH) and conventional (SFCONV) extraction of phytochemicals from tomato bagasse. To accomplish this, both flours were subjected to an in vitro stimulation of the gastrointestinal tract. After the characterization of the impact of this process on each sample, the digested samples were evaluated upon pure probiotic cultures and on fresh human fecal samples to assess the prebiotic potential and the effect on the metabolic and population dynamics of gut microflora. 

## 2. Materials and Methods

### 2.1. Tomato Bagasse Flours

#### Tomato Bagasse Flour Preparation

Two different tomato bagasse (peel and seeds) flours were used in the present work. The first (OH) was prepared from the solid by-product leftover after ohmic extraction (70 °C, 15 min, 70% ethanol). The tomato bagasse was subjected to ohmic extraction (peel and seeds) as described elsewhere [14]. The second (CONV) was prepared using the solid by-product of a conventional solid–liquid extraction described in the literature using hexane as solvent [15]. In both cases, after extraction, the leftover solid by-product fraction (SF) was dried at 55 °C overnight and stored in a desiccator at room temperature until use. The phytochemical properties of tomato flours were described in previous studies [7,10,14]. The SF for OH tomato by-products is shortened as SFOH to simplify the nomenclature, while the SF for CONV samples is SFCONV.

### 2.2. In Vitro Digestion Simulation (GID)

#### 2.2.1. Sample Preparation

The tomato SF was suspended in water (10%) and homogenized using an Ultra-Turrax (IKA Ultra-turrax T18, Wilmington, NC, USA) at 13,000× *g* for 1 min. The tomato bagasse solution was set up at 10% (*w*/*v*), as the composition showed that the dried sample contained ca. 50% fiber and, as per the European Food Safety Agency (EFSA), 6 g of fiber for each 100 g of the item [16]. Results obtained before showed the presence of 48.06 ± 0.11 g/100 g DW of insoluble dietary fiber and 46.01 ± 0.13 g/100 g DW for SFCONV [10].

#### 2.2.2. In Vitro Digestion Simulation

Before executing fecal fermentation assays, samples were subjected to an in vitro simulation of the GI tract (including dialysis) to better mimic in vivo conditions. The tomato bagasse mixture’s pH value was adjusted to 5.6–6.9, utilizing 1 M HCl. Mouth digestion was simulated by adding α-amylase from human saliva (100 U/mL in 1 mM aqueous CaCl_2_), homogenizing the mixture for 2 min, and incubating at 37 °C and 200 rpm. Afterward, to emulate stomach conditions, the mixture’s pH value was lowered to 2.0 (utilizing 1 M HCl) and pepsin from gastric juice was added (12.5 mg/mL in HCl 1 M) at a ratio of 0.05 mL/mL of sample. The mixture was then incubated in a water bath for 2 h at 37 °C and 130 rpm. To simulate small intestine conditions, the mixture’s pH was adjusted to 6.0 utilizing 1 M NaHCO_3_. Pancreatin and bile salts (0.4 g pancreatin and 1.2 g bile salts in 200 mL of NaHCO_3_ 1 M) were added to the mixture, at a ratio of 0.25 mL/mL of sample. Finally, the obtained solution was maintained at 37 °C and 45 rpm for 2 h. Afterward, the dialysis was performed with 12 kDa membranes for 24 h, with known volume (to simulate the blood circulation). At the end of the dialysis process, the solution within the dialysis tubing (OUT) represented the non-absorbable sample (colon-available). This fraction was then freeze-dried and stored in a desiccator for later use in the fecal fermentation.

### 2.3. Preliminary Evaluation of the Prebiotic Potential of Tomato SF

#### 2.3.1. Microorganisms

Probiotic bacteria species were selected for the present work, namely *Lactobacillus casei* 01 and *Bifidobacterium animalis subsp lacties* BO.

#### 2.3.2. Selection of the Best Tomato Flour Concentration

To evaluate the effect of tomato flour on the growth of the target microorganisms, tomato SF (before and after digestion) at 2, 4, and 6% (*w*/*v*) was suspended in basal media, inoculated using a 24 h inoculum at 10% (*v*/*v*), and incubated for 24 h at 37 °C in an anaerobic environment. After this period, the viable cell numbers were determined by plating, using the spread plate method, in de Mann, Rogosa, and Sharpe agar (MRS) enhanced with 0.5 g/L of L-cysteine hydrochloride. After 48 h incubation at 37 °C, under anaerobiosis, the bifidobacterial and lactobacilli colonies were enumerated, and the outcomes plotted as log CFU/mL, in accordance with [17]. All inoculations were performed in triplicate, and plain inoculated cells were determined using decimal dilutions and plating through the spread plate technique, in MRS agar enhanced with cysteine and bromophenol blue [18]. In addition, pH values were measured using a 52-02 Crison electrode, and the organic acid production was assessed through HPLC-IR, as described elsewhere [17].

### 2.4. In Vitro Fecal Fermentations

#### 2.4.1. Collection and Preparation of Fecal Inocula 

Fresh fecal samples were provided by five healthy donors (A–E, three men and two women, between the ages of 23 and 39 years old), whose selection was based on established criteria regarding health status and dietary habits; namely, to assess the existence of chronic diseases, allergies, and probiotic ingestion, among others. Moreover, an informed consent form was distributed among donors to provide the participants with information about the study, with a consent certificate assigned to each. Donors were healthy unrelated anonymous volunteers, ≥18 and <50 years of age, who had not received antibiotics in the preceding six months or consumed any prebiotic supplement. The fecal samples were maintained under anaerobic conditions for a maximum of 2 h before being used. The fecal inocula (FI) were then prepared by diluting the fecal matter in Reduced Physiological Salt solution (RPS) (constituted by 0.5 g/L cysteine-HCl (Merck, Darmstadt, Germany) and 8.5 g/L NaCl (LabChem, Zelienople, PA, USA), with a final pH value of 6.8, at 100 g/L in an anaerobic workstation (Don Whitley Scientific, West Yorkshire, UK) (10% CO_2_, 5% H_2_, and 85% N_2_) [2].

#### 2.4.2. Nutrient Base Medium Preparation

Fecal fermentations were performed with Nutrient Base Medium. The medium comprised 5.0 g/L trypticase soy broth without dextrose (Fluka Analytical, St. Louis, MO, EUA), 5.0 g/L bactopeptone (Becton Dickinson Biosciences, New Jersey, NJ, USA), 0.5 g/L cysteine-HCl (Merck, Darmstadt, Germany), 1.0% (*v*/*v*) of salt solution A [100.0 g/L NH_4_Cl (Merck, Darmstadt, Germany), 10.0 g/L MgCl_2_·6H_2_O (Merck, Darmstadt, Germany), 10.0 g/L CaCl_2_·2H_2_O (Carlo Erba, Chaussée du Vexin, France)], 1.0% (*v*/*v*) of trace mineral solution (ATCC, Manassas, VA, USA), 0.2% (*v*/*v*) of salt solution B [200.0 g/L K_2_HPO_4_·3H_2_O (Merck, Darmstadt, Germany)], and 0.2% (*v*/*v*) of a 0.5 g/L resazurin solution (Sigma-Aldrich Chemistry, St. Louis, MO, USA). The medium final pH value was adjusted to 6.8 and was then bubbled with N_2_ until it presented a translucent/yellowish color. Following this, 50 mL parts were then distributed into several containers. Fructooligosaccharides (FOS) from inulin-Raftilose^®^ P95 (Beneo-Orafti, Oreye, Belgium) with a molecular weight of 0.6 Kda—3.3 DP were used as positive controls and then freeze-dried digested [2]. Tomato residue flours were added to the respective vessels at a final concentration of 2%. The bottles were capped and autoclaved. Following sterilization, and before adding the fecal inocula, the atmosphere of each flask was refluxed with a sterile gas mixture (10% CO_2_, 5% H_2_, and 85% N_2_) [2]. 

#### 2.4.3. Fecal Fermentations

The flasks prepared before (Section 2.4.2) were inoculated at 2% (*v*/*v*) with fecal inocula (Section 2.4.1) and incubated for 48 h at 37 °C under anaerobic atmosphere (10% CO2, 5% H_2_, and 85% N_2_). Samples were collected after 0, 12, 24, and 48 h of incubation, and the pH values were measured using a MicropH 2002 pH meter (Crison, Barcelona, Spain) equipped with a 52-07 pH electrode (Crison, Barcelona, Spain). The positive and negative controls were, respectively, designated as C+ (FOS) and C- (plain media), while the digested biomass tomato flours were named OH and CONV, which are under flours. Afterward, the samples were stored at −30 °C until analysis. All the steps considered in this section were carried out inside an anaerobic workstation (Don Whitley Scientific, West Yorkshire, UK) [2].

#### 2.4.4. Fecal Fermentation Sample Processing

Aliquots of each sample (4 mL) were centrifuged for 6 min at 4000× *g*. The resulting supernatants were used to evaluate sugars and short-chain fatty acids (SCFAs), according to Section 2.6, and the pellet was used to extract the genomic DNA.

### 2.5. Sugars and SCFA Analysis

Sugar consumption and organic acid production during fecal fermentation were analyzed using an HPLC system composed of a Knauer K-1001 pump (Berlin, Germany), an ion exchange Aminex HPX87H (300 × 7.8 mm) (Bio-Rad, Hercules, CA, USA) column, and two detectors assembled in series, namely a UV-vis detector (220 nm) and a refractive index detector, both from Knauer (Berlin, Germany,) at a temperature of 65 °C. An isocratic gradient was used (13 mM H_2_SO_4_ Merck, Darmstadt, Germany), at a flow rate of 0.6 mL/min. The injection volume was 40 μL and the running time was 30 min. Fermentation supernatants were filtered through a 0.22 μm syringe filter and each sample was injected in duplicate.

### 2.6. Bacterial Population Analysis

#### 2.6.1. DNA Extraction

An NZY Tissue gDNA Isolation kit (NZYTech, Lisbon, Portugal) was used to extract DNA from the fecal samples with slight modifications. Briefly, pellets were washed with TE (pH 8.0; Tris EDTA buffer), vortexed, and centrifuged at 4000× *g* for 10 min. Then, 180 μL of a freshly prepared lysozyme solution (10 mg/mL lysozyme in a NaCl-EDTA (30 mM:10 mM) solution) was added and incubated for a period of 1 h at 37 °C, with periodic shaking. Afterward, 350 μL of NT1- buffer was added to the samples, which were then vortexed and incubated at 95 °C. After 10 min, samples were centrifuged (11,000× *g*, 10 min, 4 °C), and supernatants (200 μL) were mixed with 25 μL of proteinase K and incubated at 70 °C for 10 min. The remaining steps were performed according to the manufacturer’s instructions. After extraction, the DNA’s purity and concentration (20 ng/µL) were assessed using a Thermo Scientific™ μDrop™ Plate coupled with a Thermo Scientific™ Multiskan™ FC Microplate Photometer (Thermo Fisher Scientifc, Waltham, MA, USA). 

#### 2.6.2. Real-Time Quantitative Polymerase Chain Reaction—Gut Composition Analysis

Real-time PCR was performed using a CFX96 Touch™ Real-Time PCR Detection System (Bio-Rad Laboratories, Inc., Hercules, CA, USA), under the conditions described in Appendix A, to detect and amplify the purified bacterial gDNA [2]. The PCR reaction mixture comprised 5 μL of 2x iQTM SYBR^®^ Green Supermix (Bio-Rad Laboratories, Inc., Hercules, CA, USA), 2 μL of sterile ultrapure water, 1 μL of sample DNA (equilibrated to 20 ng/µL), and 1 μL of forward and reverse primers (100 nM) targeting the 16S rRNA gene. The primers used were obtained from STABvida (Lisbon, Portugal) and are listed in Appendix A. Standard curves were constructed using tenfold dilutions (from 2 log to 6 log of several copies of 16S rRNA gene/μL) of bacterial genomic DNA standards (DSMZ, Braunschweig, Germany); the primer sequences for qRT-PCR required in this experiment are present in Appendix A. The amplification schedule included one initial activation cycle at 95 °C for 10 min, 45 cycles at 95 °C for 10 s, an annealing step at 45, 50, or 55 °C for 60 s depending on the primer, and an extension step at 72 °C for 15 s. Melting curve analysis was performed for each PCR to evaluate the specificity of the amplification, considering a temperature interval from 60 to 97 °C, with an increase of 0.1 °C (per 0.01 min). All assays were performed in quadruplicate. Data were processed and analyzed using LightCycler software obtained from Roche Applied Science. The target groups were chosen from among the most numerous phyla and genera in the healthy human gut microbiota (Firmicutes, *Clostridium leptum* subgroup, Bacteroidetes, and Bacteroides), as well as known probiotics (*Bifidobacterium* and *Lactobacillus*). The standard curves were calculated using tenfold bacterial dilution gDNA standards of *Clostridium leptum* (ATCC 29065), *Bacteroides vulgatus* (ATCC 8482), *Bifidobacterium longum* subsp. *infantis* (ATCC 15697) (DSMZ, Braunschweig, Germany), and *Lactobacillus gasseri* (ATCC 33323) (Appendix A). The NCBI Genome database was utilized in this work to acquire the genome size and copy number of the 16S rRNA gene for each bacterial strain used as a benchmark.

### 2.7. Statistical Analysis

Statistical analysis of the data was done using IBM SPSS Statistics v21.0 (IBM, Chicago, IL, USA). The normality of the data’s distribution was evaluated through Shapiro–Wilk’s test. As the data proved to follow a normal distribution, one-way ANOVA, coupled with Tukey’s post hoc test, was used to determine the significance of the effect of tomato bagasse biomass on bacterial populations at each time point. Repeated measures ANOVA was used to evaluate the effect of tomato bagasse biomass on the bacterial population over time. Differences were considered significant for *p*-values ≤ 0.05.

## 3. Results

### 3.1. Probiotic Effect

The most used probiotic microorganisms belong to the *Lactobacillus* and *Bifidobacterium* genera. Thus, at the first stage, these microorganisms were used to understand the potential prebiotic effects of tomato bagasse flours after extraction. For this, the bacteria were inoculated into a growth medium with different concentrations of tomato flours (0%, 1%, 2%, 4%) to select the minimum concentration exerting a prebiotic effect. The results (Figure 1) showed that the flours had little impact, with differences observed between the SFCONV and SFOH samples of 2 and 4% by-product concentration. The viable cells number for *Lactobacillus* was ca. 10^8^ CFU/mL for the various percentages of tomato samples, while for *Bifidobacterium*, the observed values were lower. 

For *Lactobacillus casei*, when comparing the viable cells of the positive control and the sample at 2%, the by-product appeared to allow for some growth of this bacteria; i.e., the total viable counts were above those registered for the control and allowed for more prolonged survival of the bacterial cells, indicating that tomato biomass may be used as a source of nutrients by this microorganism. The SFOH extraction had a significant impact when compared with SFCONV (*p* < 0.05). The results suggest better bacteria accessibility to nutrients, such as carbohydrates, promoting their growth [2,18,19,20]. In addition, according to the chemical flours profile, SFOH has a greater amount of galactose, arabinose, and uronic acids than CONV, which could promote the growth of these bacteria, as described in the literature [21,22]. The results are according to the literature, given the recognized metabolic diversity of *Lactobacillus*, as previous results also reported strain-specific effects of tomato flours [23,24]. Thus, these flours can also be used as a medium for probiotic growth.

It was reported that XOS is not fermented by most of the lactobacilli tested, whereas arabinoxylan was not used by any of the strains examined [25]. However, [22] verified a *L. casei* growth in arabinoxylan. The authors also demonstrated that another strain of *Lactobacillus*, *L. brevis* DSM 20054, was genetically equipped with functional arabinoxylan-oligosaccharide-degrading hydrolases, which could explain the use of arabinose for growth.

In addition, tomato by-products could promote *L. casei* growth, suggesting that this sample may be used as a source of essential nutrients by bacteria, as has been the case in some studies that utilized tomato juice as material for the manufacture of a probiotic drink. One study showed that tomato juice enriched with *Lactobacillus plantarym ST III* strain positively affected fermented skimmed milk’s taste and health-promoting activity. A fermented tomato juice with *L. casei* and *L. plantarum* was used to create a high-bioactivity probiotic drink [26]. Phenolic compounds, lycopene, and other carotenoids are related as they cause a positive correlation between antioxidant activity and prebiotic impact [17,26,27]. No differences were found in antioxidant activity (Appendix A) between SFOH and SFCONV used in this study (87.50 ± 1.26 and 90.49 ± 2.54 g trolox eq./100 g DW, respectively), which indicated that both flours could be used as a probiotic.

Relatively to *B. animalis*, significant differences were observed at 24 h between CONV and OH extracts with 2% of tomato by-products. CONV seemed to promote the growth of this microorganism as there were viable cell numbers after 24 h incubation. Nevertheless, OH-extracted flour led to a slight decrease in bacterial growth over time. These differences between results may originate from the fact that different *Bifidobacterium* strains may have distinctive carbohydrate metabolic abilities, as has been observed in several previous studies [28,29,30,31]. Studies revealed that the *B. longum* encodes ABC transporters, PEP-PTS systems, and secondary transporters required to carry mono- and disaccharides. In comparison, *B. animalis* has a significantly smaller genome than *B. longum*, with a lower number of metabolic pathways to take advantage of carbon sources, does not encode PEP-PTS frameworks, and contains just two qualities determining sugar-specific ATP-binding proteins characteristic of ABC transporters [28]. Therefore, since previous studies [7] showed that CONV has more disaccharides and monosaccharides (glucose, fructose, and mannose) than OH (which contains more polysaccharides, such as cellulose, hemicellulose, and pectins), it is plausible that more metabolically limited bacteria, such as *B. animalis*, cannot use them as a carbon and energy source to grow.

Since there were no marginal gains in growth or the death of the target microorganism provided by the different concentrations of tomato pomace, subsequent experiments used the 2% concentration as there was limited sample availability, and in the future, it will be easy to justify as a commercial ingredient to minimize interference in final food products [32].

#### Impact of the Digested Tomato SF on Organic Acid Production 

Probiotic bacteria can produce a variety of organic acids. The primary fermentation product from the breakdown of complex dietary carbohydrates is lactic acid, specially synthesized by *Lactobacillus* and *Bifidobacterium*. In Figure 2 and Figure 3, it can be seen that at time 0 h, lactic acid is detected in the tomato SF. Greater concentrations of lactic acid concentrations were obtained, both in SFOH and in SFCONV at 2%, after 24 h. Moreover, the tomato by-product’s presence promoted an overall increase in the production/accumulation of lactic acid.

A slight increase in the production of lactic acid in *Bifidobacterium* species can be noted. This result corroborates the literature: *Bifidobacterium* produce lactic and acetic acids in large amounts, that is, larger than the amounts secreted by *Lactobacillus*, even though the latter is known to be very acid-tolerant [33].

*Bifidobacterium* and *Lactobacillus* fermentation also result in the production of acetic acid. Acetic acid was not identified in the tomato by-product control at time 0 h. Moreover, during the graphical execution of the acetic acid concentrations for the function of time, it would be expected that the concentration of this product would increase over time. However, this condition was not verified for *L. casei*, in the presence of tomato by-products, at 24 h, which presents a high standard deviation, and for the mixture of *Lactobacillus* and *Bifidobacterium*, in the absence of tomato, at 12 h. Bifidobacteria produced acetic and lactic acids at proportions of 3:2, which were not comprised by analyzing the fermentation end products: the concentration of acetic acid was approximately three times inferior to the concentration of lactic acid obtained [34]. 

*Lactobacillus* and *Bifidobacterium* can break down and metabolize a variety of substrates. The glucose concentration is initially high due to large amounts of this monosaccharide (Figure 2 and Figure 3). After 12 h, there is a decrease in its concentration since the bacterial strains consume this substrate. The metabolic capacity to consume sugars did not differ significantly between the various probiotic bacteria, except for *L. casei* in the presence of tomato by-products, which degraded glucose more sharply after 24 h, though the production of lactic acid as well as of acetic acid did not increase.

Besides glucose, the disaccharide maltose was also found in the SF but at a lower concentration. During the incubation and in SFCONV, the overall amount of maltose decreased. As bacterial enzymes hydrolyze maltose in two glucose molecules, the bacteria are likely consuming it. However, in SFOH, consumption of maltose by the microorganisms was, overall, significantly lower, with the amount of maltose in the media increasing after 12 h (possibly released from the matrix), with a subsequent reduction in the concentration after 24 h (Figure 2 and Figure 3). In SFOH, only *B animalis* was capable of consuming present maltose.

### 3.2. Impact of Tomato Flour after Extraction on Gut Microbiota

#### 3.2.1. Microbial Population Modulation

The gut microbiota assay mimics our organism’s complexity, which goes far beyond *Lactobacillus* and *Bifidobacterium*. There is a set of microorganisms that interact with each other and with different preferences for substrates. 

A simulated gut microbiota fermentation was made through an in vitro model to evaluate the potential prebiotic impact of tomato flours (promotion of positive microorganism growth and metabolite production) obtained after CONV and OH extraction. 

The phyla Bacteroidetes, which are Gram-negative, and Firmicutes, which are Gram-positive, are the most plentiful in the human gut. Bacteroidetes and Bacteroides presented significant differences between SFCONV and the controls. For the Bacteroidetes cluster, there was an increase in the presence of SFCONV at 12 h, with significant differences compared with C^-^. In addition, according to Figure 4, there is a greater dispersion in the number of gene copies for SFOH than for SFCONV. The latter population is less spread out, concentrating in the 5 log of number of copies of 16S rRNA/ng DNA). It is also possible to verify that for SFOH, about 25% of the population presented a 6 log 16S rRNA gene copies/ng of DNA, similar in behavior to FOS. The results are in agreement with the literature, as Bacteroidetes may metabolize complex nutrient polymers, many of which are molecules in the plant cell wall (e.g., cellulose, pectin, and xylan), which, through the of action human digestive enzymes’ cleavage activity, are released and may reach the colon intact [35]. Studies reveal that dietary habits and lifestyle turn into determinants and play a critical part in gut microbiota variations. High-fiber and animal protein foods increase Bacteroidetes, whereas the presence of high-fiber and carbohydrate foods increases Firmicutes and Prevotella [36,37]. This information reinforces our results since SFOH presents more protein and insoluble fiber than SFCONV and, consequently, more Bacteroidetes than SFOH (*p* < 0.05) [10].

In addition, SFCONV samples contain more bound phenolics than SFOH. Xue and colleagues (2016) showed that phenolic compounds, namely quercetin and catechin, inhibit the growth of Bacteroidetes and Firmicutes. Nevertheless, other microorganisms maintain the ability for carbohydrate and energy metabolism in each group. It is still unknown how other bacteria use FOS and their metabolites [35]. The presence of multiple FOS transport systems with different specificities in each strain may also explain the selective metabolism of particular oligosaccharide components observed here. Moreover, bound phenolic compounds may be responsible for altering the metabolism pathway, inhibiting Bacteroidetes growth. The increase of Bacteroidetes leads to an increase of acidic compounds such as pyruvic, citric, fumaric, and malic acids, indicators of higher energy metabolism, and thus contributes to the healthy metabolome [38,39,40]. For Bacteroides, the results also showed the same tendency as with Bacteroidetes genera. SFOH presented a more heterogeneous distribution for Bacteroides than SFCONV, which presents similar behavior for different donors (Figure 4).

Furthermore, the results showed that 25% of SFOH and SFCONV have a gene copy number higher than FOS. Additionally, there is a significant increase in the Bacteroides population caused by SFOH and SFCONV at 12 h when compared to controls (*p* < 0.05) (Figure 2). The SFCONV have more rutin than SFOH, a polyphenol compound, as described in [7]. The authors claimed that polyphenols might modify the microbiota balance through biased effects on Bacteroides [41]. 

Regarding the Firmicutes results, a slight increase of 16S rRNA at 24 h for SFCONV samples’ exposure was observed compared to control samples, with significant differences (*p* < 0.05). According to Figure 2, while the SFOH samples presented similar results to FOS, the bacterial population is more dispersed than the observed SFCONV bacterial population. SFOH contains more fatty acids and fiber than SFCONV (Appendix A). Studies have demonstrated that a diet rich in fiber and low in fat promotes Firmicutes’ growth, which metabolizes dietary plant-derived polysaccharides to SCFAs [36,42]. 

The ratio of Firmicutes to Bacteroidetes (F/B) was also analyzed at 1:1 for all samples during the experience. Commonly, healthy individuals display a nearly 1:1 ratio of F/B [35,43], and the ratio’s increase (e.g., to 20:1, F/B) or decrease has been associated with obesity and weight loss, respectively [44]. In addition, dietary intake, such as fiber, and phytochemicals have a higher impact on microbiota. An example is a diet rich in fiber, which increases Bacteroidetes; diets rich in calories increase Firmicutes. The maintenance of the F/B ratio during all experiences corroborates the higher amounts of fatty acids and dietary fiber present in samples, contributing to the ratio’s equilibrium. Thus, this is a good indicator of the use of SFOH in diets to contribute to the health of individuals. Other studies with obese or malnourished individuals could be interesting to understand the alterations caused by these samples.

In general, the *Bifidobacterium* showed a slight increase over time for both tomato SFs tested. According to Figure 4, about 50% of the initial population of *Bifidobacterium* is between 1 and 4 log of copies of the number of 16S rRNA/ng DNA. At 6 h fermentation, a significant increase was observed in the number of copies with time (*p* < 0.05) for SFOH. In addition, CONV fermentations showed higher levels of gene copies at 12 h compared to the controls. In addition, at 24 h, 25% of the bacteria presented more copies than FOS. Given the results showed by [7], it appears that SFCONV has more soluble fiber than SFOH and FOS, being a good carbon source and promoting a greater growth of this bacteria. Studies have shown that, depending on the strains, they use different substrates for growth. While not all strains are capable of using (most of the components of) galactooligosaccharides, the capacity of intestinal communities for the metabolization of galactooligosaccharides would not be excluded [45]. The combined activity of multiple bacteria is responsible for the fermentation of complex carbohydrates in the gut [21]. Researchers have investigated several strains and revealed that 11 of the bifidobacterial strains were significantly growing on polydextrose (soluble fiber), final OD_600_ > 05, whereas 34 of the bifidobacterial strains exhibited positive growth FOS. This is an essential result to deduce the prebiotic potential of different carbohydrates to increase the diversity of the gut microbiota. McLaughlin et al. (2015) verified superior growth with inulin to *B. longum* subsp. CCUG 18157 when compared to the other strains tested. It is possible that this strain produces a specific enzyme, such as β-fructofuranosidase, with specificity for FOS or inulin.

One intestinal bacterium naturally existing in the gut microbiota of healthy people is *Akkermansia.* When present in the intestinal flora, this group also produces propionate and acetate; however, the fecal samples contained lower gene copies of *Akkermansia.* The fermentation of both samples induced a significant reduction in *Akkermansia* levels from 0 to 12 h. Nonetheless, no differences were found between samples and controls (*p* > 0.05). The lower concentrations of this bacteria, when compared with other groups of bacteria, could be associated with its human intestinal colonization at a very young age (it is found in breast milk and infant formula) (Lukovac et al., 2014); the median age of the donors was 40 years old. Additionally, recent studies have shown that diets rich in fiber and protein decrease the *Akkermansia* population [46,47]. Our results agree with previous reports, where SFOH presents more fat and soluble fiber than SFCONV, resulting in a population distribution with less 16 rRNA gene copies of *Akkermansia*.

*C. leptum* belongs to the group of anaerobic bacteria that mainly produce propionate and butyrate in gut microbiota and use amino acids as the primary energy source. No differences were observed in cell numbers (*p* > 0.05); nevertheless, interesting results were observed in Figure 4. The first quartile population (25%) of SFOH presented a lower number of gene copies than other samples, while the second and third quartiles, 50% of the *Clostridium leptum* population, presented similar 16 rRNA gene copies with FOS. Regarding SFCONV, this seems to promote the growth of this bacteria better. As seen with *Akkermansia*, a diet rich in fermentable fiber can promote the growth of clostridium; however, the presence of lipids can also inhibit it, thus verifying the discrepancies in the growth of this microorganism [36,48]. 

The differences obtained for microorganisms agree with recent research, suggesting that food changes may drastically modify endogenous microbial communities’ total composition and organization in the gut.

#### 3.2.2. SCFA Analysis

As referred to previously, some of the welfare benefits attributed to fiber fermentation by the colonic bacteria are related to the metabolites generated. 

Relative to butyrate, SCFAs (Figure 5) are the primary energy source for normal, healthy colon cells. In addition, they safeguard against colon cancer and inflammation due to their capacity to help defend the gene-expression structure that discourages the development and proliferation of cancer cells. Results suggest a significant increase of n-butyrate at 6 h in samples fermented with SFOH- compared with control samples. At 24 h, the SFCONV-fermented samples also showed an increase of n-butyrate higher than SFOH samples, but no statistical differences were found (*p* > 0.05).

The results showed a significantly higher acetate concentration for tomato flour CONV than for OH (*p* < 0.05). In addition, there was an increase in acetate concentration over time. The results align with previously reported observations since SCFAs are, for the most part, created by enteric microorganisms as Bacteroidetes and *Bifidobacterium* because of carbohydrate fermentation through the hydrolysis of acetyl-CoA. A little part is synthesized by acetogenic microorganisms that use hydrogen, carbon dioxide, or formic acid through the Wood–Ljungdahl pathway [49,50]. The results are aligned with the literature, since the observed formic acid concentration decreases as acetate concentration increases.

There was a slight decrease in propionate concentration at 12 h in the SFOH sample compared with the positive control, with a significant difference (*p* < 0.05). Nevertheless, increased propionate concentration was found at 24 h for both positive control and tomato bagasse flours (SFOH, SFCONV), with significant differences compared to the negative control. The different propionate pathways may explain these results: succinate, acrylate, and propanediol. The succinate pathway is related to the Firmicutes and Bacteroidetes [50,51]. The results are illustrated in Figure 5, where succinate concentration decreases over time, suggesting a production of propionate based on the succinate pathway.

The results showed higher production of butyrate at 24 h for SFOH and SFCONV samples than C-, which indicates a tomato bagasse flour fermentation stimulating the production of this acid.

Although acetate and propionate production for SFOH and SFCONV were almost the same, the CONV sample had higher butyrate production than the OH sample. In addition, acetate and propionate are related to the advancement of satiety, thus taking into account the phytochemical profile of the tomato bagasse flour and the results obtained for propionate and acetate production; they could be applied as a substitute for animal-derived proteins and fiber in foods.

According to the Pearson correlation (Figure 6), a significant impact is observed in some SCFAs on the expression of some microorganisms, which is considered in the evaluation of the previously described results. Propionate is correlated with the growth of Clostridium and Firmicutes, and the production of formate and succinate is correlated with Akkermancia. The last one also influences the Firmicutes.

## 4. Conclusions

The screening of the prebiotic properties of SF obtained after OH and CONV extraction from tomato by-products was assessed using *Lactobacillus* and *Bifidobacterium* as probiotics. Differences in bacterial carbohydrate utilization patterns between species were identified, with the best results being obtained for Bifidobacterium animalis BO. The impact of SFOH and SFCONV on these probiotics was small, with differences observed for the *Lactobacillus* and Bifidobacteria strains. SFOH at 2% and 4% contributes to *L. casei* growth when compared to SFCONV, while for Bifidobacterium, SFCONV at 2% promotes it growth.

Regarding the fecal fermentation based on volunteers’ feces, both flours’ main groups are the *Bifidobacterium* and *Akkermansia*. In addition, 25% of SFOH and SFCONV samples presented gene copies higher than the positive control for Bacteroides. Regarding SFOH, this sample enhanced Bacteroidetes’ growth. In addition, 50% of population tests presented similar results with FOS to *C. leptum*, while SFCONV presented a higher number of genes copies than the positive control for *Bifidobacterium*.

Concerning SCFA results, both flours increased the propionate, butyrate, and acetate concentration compared to the negative control, which indicates the production capacity of these acids by SFOH and SFCONV during fermentation. In addition, SFCONV produces more butyrate than the OH samples.

However, a relation was observed between certain bacterial groups and SCFA concentration. For the *Bifidobacterium*, both acetate and n-butyrate influence its growth, while *Clostridium* is influenced by iso-butyrate and propionate.

The outcomes propose that both tomato flours favor a potential modulatory impact upon the gut microbiota, thus giving a counteractive action for different diets. Moreover, SFOH comes from a cleaner extraction than SFCONV, making it a potential sustainable ingredient with a prebiotic impact through the growth enhancement of *Bifidobacterium animalis* and improvement of the generation of SCFA.

Nonetheless, initial and promising evidence of their potential prebiotic effect was demonstrated, raising the need for more extensive testing in vivo as part of future work.

## Figures and Tables

**Figure 1 foods-12-01920-f001:**
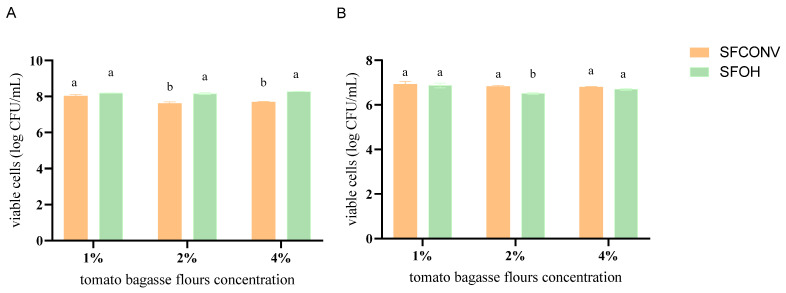
Impact of different concentrations of digested tomato by-products (after carotene extraction, OH, and CONV) on the growth of Lactobacillus (**A**) and *Bifidobacterium* (**B**) after 24 h anaerobic incubation. Letters mean the significant difference between methods for each tomato flours’ concentration *p* < 0.05.

**Figure 2 foods-12-01920-f002:**
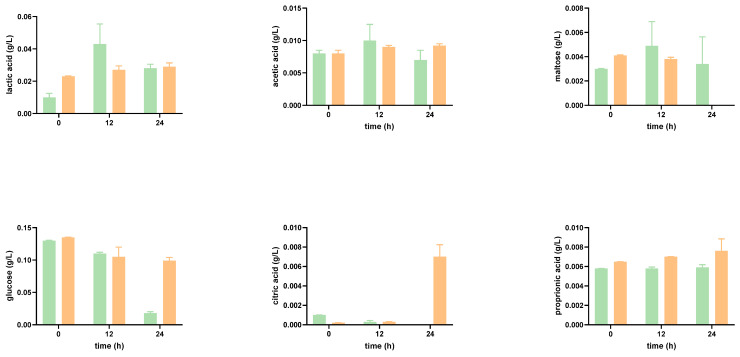
Concentrations of organic acids and sugars during the 24 h of growth of prebiotic bacteria Lactobacillus, incubated in the presence of SFOH (green) and SFCONV (orange) of tomato SF 2%, after simulation of the gastrointestinal tract.

**Figure 3 foods-12-01920-f003:**
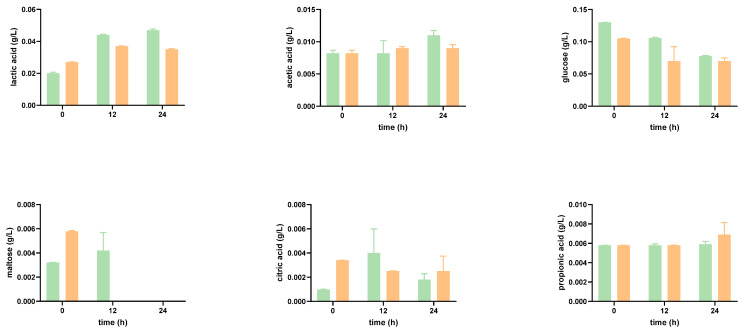
Concentrations of organic acids and sugars during the 24 h of growth of prebiotic bacteria *Bifidobacterium*, incubated in the presence of SFOH (green) and SFCONV (orange) of tomato SF 2%, after simulation of the gastrointestinal tract. Citric acid, in the control, increased throughout the incubation time for all probiotic bacteria as well as in the mixture of prebiotics. In addition, a significant concentration of citric acid was produced by *L. casei*. Towards the presence of tomato by-products, an increase in citric acid is visible in the first 12 h; however, after this time, the concentration decreases substantially, with *L. casei* production reaching null values. It is possible to conclude that *B. animalis* and *B. longum* achieved the highest concentrations in the presence of these flours.

**Figure 4 foods-12-01920-f004:**
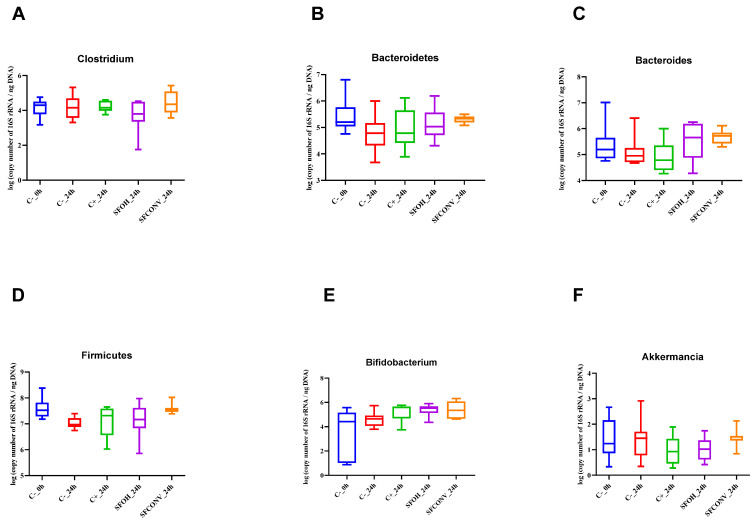
Distribution of gut bacterial populations (log 16S rRNA gene copies/ng of DNA, means ± SD) detected by PCR in fecal samples. The used probes: *Clostridium leptum* (**A**), Bacteroidetes (**B**), Bacteroides (**C**), Firmicutes (**D**), *Bifidobacterium* (**E**), and *Akkermansia* (**F**).

**Figure 5 foods-12-01920-f005:**
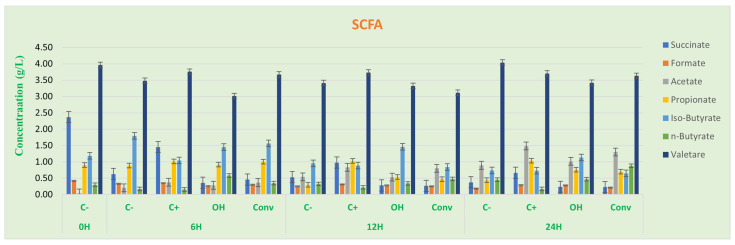
Concentration (mg/mL ± SD) of the SCFAs produced along with fermentation time in fecal samples. Negative control (C-), positive control (C+), tomato residue flour after OH extraction (OH), and tomato residue flour after conventional extraction (CONV). Different letters mark statistically significant (*p* < 0.05) differences.

**Figure 6 foods-12-01920-f006:**
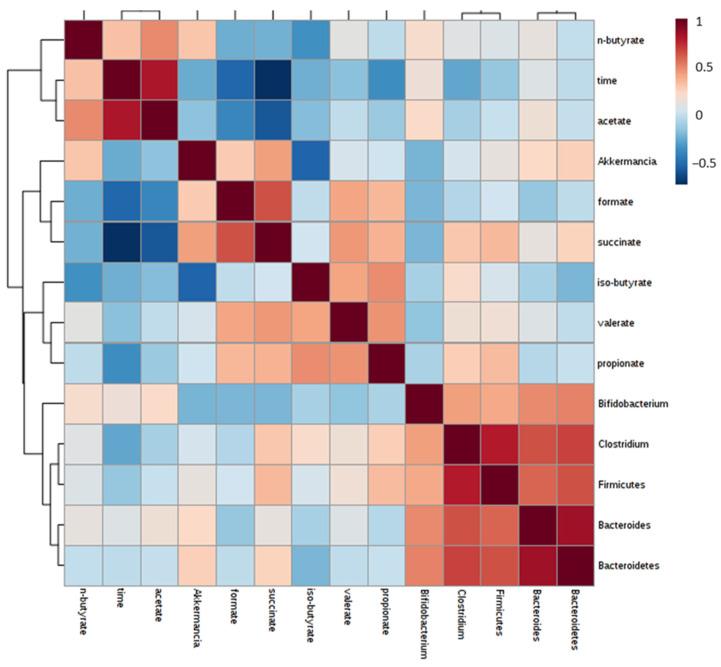
Pearson correlation between microorganisms and SCFAs produced during fermentations.

## Data Availability

The data presented in this study are available on request from the corresponding author. The data are not publicly available due to restrictions, e.g., privacy or ethics.

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
