# Peer review of "Modulation of the Gut Microbiota by Tomato Flours Obtained after Conventional and Ohmic Heating Extraction and Its Prebiotic Propertiesâ€"

_foods, 2023, doi:10.3390/foods12091920_

Round 1
Reviewer 1 Report
The positive functional health effects of both prebiotics and pro-biotics on gut microbiota was reviewed. Among these, the selective growth of beneficial bacteria due to the use of prebiotics and bioactive compounds as an energy and carbon source is critical to promote the development of healthy microbiota within the human gut. The present work aimed to assess the fermentability of tomato flour obtained after ohmic (SFOH) and conventional (SFCONV) extraction of phenolic compounds and carotenoids as well as their potential impact upon specific microbiota groups. To accomplish this, the attained bagasse flour was submitted to an in vitro simulation of gastrointestinal digestion before its potential fermentability and impact upon gut microbiota (using an in vitro fecal fermentation model). Different impact on the probiotic strains studied was observed for SFCONV promoting the B. animalis growth, while SFOH promoted the B. longum, probably based on the different carbohydrate profiles of those flours. Overall, the flours used were capable of functioning as a direct substrate to support the potential prebiotic growth for Bifidus longum.
The fecal fermentation model results showed the highest Bacteroidetes growth with SFOH and the highest values of Bacteroides with SFCONV. A correlation between microorganisms' growth and short-fatty acids was also found. This by-product seems to promote beneficial effects on microbiota flora and could be a potential prebiotic ingredient, although more extensive in vivo trials would be necessary to confirm this.
It is interesting topic and the manuscript is well organized. However, there still have some issues need to check. The reference should be updated.
1. The introduction should be compressed and rearranged.
2. The tomato flour obtained after ohmic (SFOH) and conventional (SFCONV) extraction of phenolic compounds and carotenoids should be measured.
3. “2.4 Preliminary evaluation of the prebiotic potential of tomato SF” should refer the reference (Food & Function, 2022, 13(24), 12686-12696. Doi: 10.1039/d2fo01746f).
4. “3.1. Probiotic effect”. The most used probiotic microorganisms belong to the Lactobacillus and Bifidobacterium genera. The probiotic effect should combine with the antioxidative effect (Food Chemistry, 402(2023): 134231).
5. The discussion should be compressed.
6. The gut microbiota information of 16sR DNA should be supplement, which may be consist with the dose and processing technoloy.
Author Response
Dear Reviewer,
First, the authors sincerely acknowledge the interest demonstrated in our work and the availability to reconsider a revised version of this manuscript.
We want to thank all the positive inputs and suggestions given by the reviewer, which contribute to improving and enriching this manuscript.
The answers are given just after the transcription of your comments, and new information is added to the article with tracked changes as requested in the revised version.
- The introduction should be compressed and rearranged.
- The introduction was improved accordingly.
- The tomato flour obtained after ohmic (SFOH) and conventional (SFCONV) extraction of phenolic compounds and carotenoids should be measured.
- The tomato flour was characterized and published in (Coelho et al., 2021, 2023).
- “2.4 Preliminary evaluation of the prebiotic potential of tomato SF” should refer the reference (Food & Function, 2022, 13(24), 12686-12696. Doi: 10.1039/d2fo01746f).
- The reference was included.
- “3.1. Probiotic effect”. The most used probiotic microorganisms belong to the Lactobacillus and Bifidobacterium genera. The probiotic effect should combine with the antioxidative effect (Food Chemistry, 402(2023): 134231).
- The discussion was improved and the reference added.
- The discussion should be compressed.
- The discussion was compressed.
- The gut microbiota information of 16sR DNA should be supplement, which may be consist with the dose and processing technoloy.
- The information was put as supplementary material.
References
Coelho, M. C., Ghalamara, S., Campos, D., Ribeiro, T. B., Pereira, R., Rodrigues, A. S., Teixeira, J. A., & Pintado, M. (2023). Tomato Processing By-Products Valorisation through Ohmic Heating Approach. Foods, 12(4), 818. https://doi.org/10.3390/foods12040818
Coelho, M. C., Ribeiro, T. B., Oliveira, C., Batista, P., Castro, P., Monforte, A. R., Rodrigues, A. S., Teixeira, J., & Pintado, M. (2021). In Vitro Gastrointestinal Digestion Impact on the Bioaccessibility and Antioxidant Capacity of Bioactive Compounds from Tomato Flours Obtained after Conventional and Ohmic Heating Extraction. In Foods (Vol. 10, Issue 3). https://doi.org/10.3390/foods10030554

Reviewer 2 Report
The manuscript entitled “Flours from tomato bagasse obtained after conventional and ohmic heating extraction: impact of digestion and prebiotic effect” aims to assess the prebiotic potential of two tomato flours obtained after two different extraction procedures for phytochemicals, namely ohmic and conventional. To this aim Lactobacillus casei and Bifidobacterium were used as probiotic model microrganisms.
Generally speaking the manuscript is well written, experiments well designed and results well discussed and proven by consistent statistical analysis. The study was presented in an interesting way and highlights that tomatoe by-products, such as bagasse, could be re-used to obtain nutritive flours to be used as ingredient in foods because their prebiotic potential. This sound very well from a circulary economy perspective.
In summary I have no substantive comments on the content of the publication, because in my opinion all aspects have been covered there. These are only few comments:
Introduction line 76-79: please the authors rephrased the sentence “A full and integrated recov-76 ery from tomato by-products with zero residues, in a context of a circular economy, the 77 final solid extraction by-product can be dried under controlled conditions resulting in 78 flour with a high fiber content combined with bond phytochemicals” because it is not very clear.
Materials and method, par. 2.1.1. Tomatoe bagasse flour preparation. I would suggest to add a figure reporting the workflow for the samples preparation for clarity purpose.
I suggest some minor revisions.
Author Response
Dear Reviewer,
First, the authors sincerely acknowledge the interest demonstrated in our work and the availability to reconsider a revised version of this manuscript.
We want to thank all the positive inputs and suggestions given by the reviewer, which contribute to improving and enriching this manuscript.
The answers are given just after the transcription of your comments, and new information is added to the article with tracked changes as requested in the revised version:
"In summary I have no substantive comments on the content of the publication, because in my opinion all aspects have been covered there. These are only few comments:
Introduction line 76-79: please the authors rephrased the sentence “A full and integrated recov-76 ery from tomato by-products with zero residues, in a context of a circular economy, the 77 final solid extraction by-product can be dried under controlled conditions resulting in 78 flour with a high fiber content combined with bond phytochemicals” because it is not very clear.
R. The sentence was improved.
Materials and method, par. 2.1.1. Tomatoe bagasse flour preparation. I would suggest to add a figure reporting the workflow for the samples preparation for clarity purpose.
R. A graphical abstract was added to clarify the purpose (the manuscript has many figures).

Reviewer 3 Report
The article entitled of Flours from tomato bagasse obtained after conventional and 2 ohmic heating extraction: impact of digestion and prebiotic effect was an interested topic for readers of the journal of Foods. However, there were a few points that needed to be concenred and clarified as following.
1. In the introduction and results section, there was no information of the physio chemical properties of the substrates used in the experiments.
2. The authors assumed that Tomato by-products include seeds, peels, and pulp, all of which are rich in nutrients and bioactive, including carbohydrates, organic acids, pigments, fiber, proteins, oils, and vitamins giving beneficial effects on health but there was no proved in the details that the substrates from different parts used in this study.
3. Molecular weight of the substrated have to be informed in the results section.
Author Response
The article entitled of Flours from tomato bagasse obtained after conventional and 2 ohmic heating extraction: impact of digestion and prebiotic effect was an interested topic for readers of the journal of Foods. However, there were a few points that needed to be concenred and clarified as following.
In the introduction and results section, there was no information of the physio chemical properties of the substrates used in the experiments.
R. The introduction and results were improved and references to data was also included.
The authors assumed that Tomato by-products include seeds, peels, and pulp, all of which are rich in nutrients and bioactive, including carbohydrates, organic acids, pigments, fiber, proteins, oils, and vitamins giving beneficial effects on health but there was no proved in the details that the substrates from different parts used in this study.
R. The data with phytochemical properties is now published and included in the paper
Molecular weight of the substrated have to be informed in the results section.
R. The results were improved. Fructooligosaccharides (FOS) inulin-Raftilose® P95 (Beneo-Orafti, Belgium) with Molecular weight of 0.6 Kda – 3.3 DP was used as positive control and be well characterized in study performed before [1]. Tomato flours are also characterized by Coelho and coleagues [2] and the information were added in the results.
[1] Roupar, D.; Coelho, M.C.; Gonçalves, D.A.; Silva, S.P.; Coelho, E.; Silva, S.; Coimbra, M.A.; Pintado, M.; Teixeira, J.A.; Nobre, C. Evaluation of Microbial-Fructo-Oligosaccharides Metabolism by Human Gut Microbiota Fermentation as Compared to Commercial Inulin-Derived Oligosaccharides. Foods 2022, 11, 954, doi:10.3390/foods11070954.
[2] Coelho, M. C., Ghalamara, S., Campos, D., Ribeiro, T. B., Pereira, R., Rodrigues, A. S., Teixeira, J. A., & Pintado, M. (2023). Tomato Processing By-Products Valorisation through Ohmic Heating Approach. Foods, 12(4), 818. https://doi.org/10.3390/foods12040818

Round 2
Reviewer 1 Report
Several studies have supported the positive functional health effects of both prebiotics and pro-biotics on gut microbiota. Among these, the selective growth of beneficial bacteria due to the use of prebiotics and bioactive compounds as an energy and carbon source is critical to promote the development of healthy microbiota within the human gut. The present work aimed to assess the fermentability of tomato flour obtained after ohmic (SFOH) and conventional (SFCONV) extrac-tion of phenolic compounds and carotenoids as well as their potential impact upon specific mi-crobiota groups. To accomplish this, the attained bagasse flour was submitted to an in vitro sim-ulation of gastrointestinal digestion before its potential fermentability and impact upon gut mi-crobiota (using an in vitro fecal fermentation model). Different impact on the probiotic strains studied was observed for SFCONV promoting the B. animalis growth, while SFOH promoted the B. longum, probably based on the different carbohydrate profiles of those flours. Overall, the flours used were capable of functioning as a direct substrate to support the potential prebiotic growth for Bifidus longum. The fecal fermentation model results showed the highest Bacteroidetes growth with SFOH and the highest values of Bacteroides with SFCONV. A correlation between microorganisms' growth and short-fatty acids was also found. This by-product seems to promote beneficial effects on microbiota flora and could be a potential prebiotic ingredient, although more extensive in vivo trials would be necessary to confirm this.
It is interesting topic and the manuscript is well organized. However, there still have some issues need to check.
The reference should be updated:
1. The introduction should be compressed and rearranged.
2. The tomato flour obtained after ohmic (SFOH) and conventional (SFCONV) extraction of phenolic compounds and carotenoids should be measured.
3. “2.4 Preliminary evaluation of the prebiotic
potential of tomato SF” should refer the reference
(Food & Function, 2022, 13(24), 12686-12696. Doi: 10.1039/d2fo01746f).
4. “3.1. Probiotic effect”. The most used probiotic microorganisms belong to the Lactobacillus and Bifidobacterium genera. The probiotic effect should combine with the antioxidative effect.
5. The discussion should be compressed.
6. The gut microbiota information of 16sR DNA should be supplement, which may be consist with the dose and processing technology (Food Bioscience. 50(2022):101946. Doi: 10.1016/j.fbio.2022.101946)."
Author Response
Dear Reviewer,
First, the authors sincerely acknowledge the interest demonstrated in our work and the availability to reconsider a revised version of this manuscript.
We want to thank all the positive inputs and suggestions given by the reviewer, which contribute to improving and enriching this manuscript.
The answers are given just after the transcription of your comments, and new information is added to the article with tracked changes as requested in the revised version.
The reference should be updated:
R. The references were improved.
The introduction should be compressed and rearranged.
R. The introduction was compressed and rearranged.
The tomato flour obtained after ohmic (SFOH) and conventional (SFCONV) extraction of phenolic compounds and carotenoids should be measured.
R. The tomato flour was measured and published in (Coelho et al., 2021, and Coelho et al. 2023), references were included in the material and methods.
“2.4 Preliminary evaluation of the prebiotic potential of tomato SF” should refer the reference (Food & Function, 2022, 13(24), 12686-12696. Doi: 10.1039/d2fo01746f).
R. The reference was included.
“3.1. Probiotic effect”. The most used probiotic microorganisms belong to the Lactobacillus and Bifidobacterium genera. The probiotic effect should combine with the antioxidative effect.
R. The probiotic was combined and the information is in supplementary material.
The discussion should be compressed.
R. The discussion was compressed.
The gut microbiota information of 16sR DNA should be supplement, which may be consist with the dose and processing technology (Food Bioscience. 50(2022):101946. Doi: 10.1016/j.fbio.2022.101946).
R. The methodology was improved.
Reviewer 3 Report
Figure 2, 6 and 7 needed to be corrected.
Also, deep discussion of prebiotics influenced the gut microbiome needed to be compared in term of FOS and so on. Any types of microorganisms were different. the authors needed to be discussed.
Author Response
Dear Reviewer,
First, the authors sincerely acknowledge the interest demonstrated in our work and the availability to reconsider a revised version of this manuscript.
We want to thank all the positive inputs and suggestions given by the reviewer, which contribute to improving and enriching this manuscript.
The answers are given just after the transcription of your comments, and new information is added to the article with tracked changes as requested in the revised version.
Figure 2, 6 and 7 needed to be corrected.
R. The figures were improved.
Also, deep discussion of prebiotics influenced the gut microbiome needed to be compared in term of FOS and so on. Any types of microorganisms were different. the authors needed to be discussed.
R. The discussion was improved.